# Moody Mares—Is Ovariectomy a Solution?

**DOI:** 10.3390/ani10071210

**Published:** 2020-07-16

**Authors:** Daniel Taasti Melgaard, Trine Stokbro Korsgaard, Martin Soendergaard Thoefner, Morten Roenn Petersen, Hanne Gervi Pedersen

**Affiliations:** 1Hoersholm Equine Clinic, Kongevejen 124D, 3480 Fredensborg, Denmark; DM@hestepraksis.com (D.T.M.); MT@hestepraksis.com (M.S.T.); 2DyrDoktor, Sjaellandsgade 20, 6400 Soenderborg, Denmark; trineskorsgaard@googlemail.com; 3Section for Veterinary Reproduction and Obstetrics, Department of Veterinary Clinical Sciences, University of Copenhagen, Højbakkegård Allé 5A, 2630 Taastrup, Denmark; mortenpetersenr@gmail.com

**Keywords:** moody mares, ovariectomy, ovarian neoplasia, tumour, unwanted behaviour

## Abstract

**Simple Summary:**

Unwanted behaviour in mares is a commonly presenting problem to the veterinarian. This behaviour may range from the mare being uncooperative or aggressive when handled on the ground, kicking, bucking or rearing when ridden or being aggressive towards other horses. In some cases, ovarian neoplasms (cancers) cause the mare to change behaviour, but in other cases, there is no apparent reason for the unwanted behaviour. The aim of the present study was to evaluate whether ovarian removal in mares with unexplained unwanted behaviour improved the mare’s behaviour or rideability from the owner’s perspective. Twenty-eight mares underwent surgical ovarian removal. Fourteen mares had ovarian neoplasia in either one or both ovaries, 10 mares had normal ovaries, and the ovaries of the remaining four mares were not examined to identify if cancer was present. Following ovariectomy, rideability improved in 80% (8/10) of mares with normal ovaries and in 57% (8/14) of mares with ovarian neoplasm. A behavioural improvement was observed in 40% (4/10) of mares with normal ovaries, and in 43% (6/14) of mares with ovarian neoplasm. Mares with unwanted behaviour not obviously related to the oestrus cycle and to painful conditions may benefit from ovariectomy to alter their behaviour and rideability.

**Abstract:**

Unwanted behaviour in mares is a commonly presenting problem to the veterinarian. This behaviour may range from the mare being uncooperative or aggressive when handled on the ground, kicking, bucking or rearing when ridden or being aggressive towards other horses. This purpose of the study was to evaluate whether bilateral ovariectomy in mares with unwanted behaviour improved the mare’s behaviour and/or rideability from the owner’s perspective. The mares were grouped and compared statistically based on their histological classification as having either “normal” or neoplastic ovaries. This study is a retrospective case series report of twenty-eight ovariectomized mares. A semi-quantitative value (1–10) pre- and post-ovariectomy for A) behaviour on ground/in stable and B) rideability was given, based on the owner’s observations. The horses were grouped based on their histopathologic diagnosis as “Normal ovaries” or “Neoplasia”. Following ovariectomy, rideability improved, with a score of ≥5 in 80% (8/10) of mares with normal ovaries and in 57% (8/14) of mares with ovarian neoplasm. A behavioural improvement of ≥5 was observed in 40% (4/10) of mares with normal ovaries, and in 43% (6/14) of mares with ovarian neoplasm. A significant difference was observed between the semi-quantitative value pre- and post-ovariectomy in both groups. No difference was observed in change in behaviour and rideability score between the group with normal ovaries and neoplastic ovaries. Mares with unwanted behaviour not obviously related to the oestrus cycle and to painful conditions may benefit from ovariectomy to alter their behaviour and rideability.

## 1. Introduction

Unwanted behaviour in mares is a commonly presenting problem to the veterinarian. The unwanted behaviour can be normal oestrous behaviour, which causes the mare to be less cooperative or to perform worse during oestrus. In other cases, the behaviour cannot be directly related to the oestrus cycle. This behaviour may range from the mare being uncooperative or aggressive when handled on the ground, kicking, bucking or rearing when ridden or being aggressive towards other horses. This kind of unwanted behaviour may stem from medical conditions causing pain or underlying disease, e.g., granulosa-theca cell tumours causing the mares to show aggressive stallion-like behaviour. It is a diagnostic challenge to discern between the origins of the aberrant behaviour. If no obvious pathology of the ovaries exists, the behaviour may be due to referred pain from the vagina or bladder, visceral, gastric, dental or musculoskeletal pain. If the owner perceives that unwanted behaviour is related to the oestrus cycle, the mare can be treated with altrenogest, a GnRH vaccine or other methods of postponing oestrus to assess whether the unwanted behaviour disappears. Bilateral ovariectomy (OVE) has been used to correct behavioural problems related to the oestrus cycle. Following OVE, aggressive behaviour associated with the oestrous cycle was resolved in three out of four mares [1], while OVE improved the behaviour of nine out of nine mares with unruly behaviour and decreased athletic performance ascribed to oestrus [2]. Ten out of 12 (83%) owners of mares with objectional oestrus behaviour found that the mares performed better after OVE due to the elimination of fluctuations in performance [3]. Kamm and Hendrickson (2007) [4] assessed owner perception of improvement in mare behaviour following bilateral OVE and reported that 83% (19/23) of owners perceived an improvement in mare behaviour, aggression problems improved in 86% (12/14), kicking and biting at other horses improved in 73% (8/11), training problems improved in 72% (13/18) and problems with other horses improved in 64% (9/14) of the cases. In the study by Kamm and Hendrickson (2007), it was not specified if the unwanted behaviour was oestrus-related.

Despite a full diagnostic work-up to locate the source of the unwanted behaviour, some mares are still without a diagnosis that can explain their behaviour. The mares participating in the study were a subset of mares that had already been screened for painful conditions stemming from outside the reproductive tract, but included mares with ovarian tumours. OVE was used as a last-ditch attempt to solve the behavioural problems in mares in which pharmacological treatment to abolish cyclicity did not alter the behaviour and where no other problem outside the reproductive tract was localized.

### Aim

The aim of the present study was to evaluate the effect of bilateral OVE in mares with unexplained unwanted behaviour from the owner’s perspective regarding the mare’s behaviour and rideability.

Further, we hypothesized that mares with ovarian neoplasia would have a more disagreeable demeanour pre-OVE than mares without neoplasia, and therefore the former group would have a higher success rate of regaining normal behaviour after OVE.

## 2. Materials and Methods

The design of the study is a retrospective case series report of ovariectomized mares. All owners gave their informed consent for inclusion before their mare participated in the study.

Twenty-eight mares exhibiting rideability and/or behavioural problems were selected for OVE. The dataset for each mare consisted of a medical record, an owner questionnaire (Appendix A) and a telephone interview. The mares were aged between 5 and 18 years and of Warmblooded breeds e.g., Danish Warmblood, Hanoverian and KWPN. The mares were used for dressage and show jumping. Experience level was not recorded. The data collection was conducted a minimum of 6 and a maximum of 24 months after the time of surgery. The questionnaire and telephone interview depicted the owner’s subjective view of the performance and behaviour pre- and post-OVE. A semi-quantitative value (1–10) pre- and post-OVE for (1) behaviour and (2) rideability was given, based on the owners’ observation. The value 1 was defined as dangerous to ride or handle or aggressive towards other horses. The value 10 was the best category, which corresponded to a mare being very rideable, easy to handle and showing no aggressive behaviour towards other horses. Behaviour was defined as problems experienced on the ground.

Inclusion criteria for OVE were either (1) a thorough work-up identifying no extra-ovarian primary pathology or (2) ovarian neoplasia detected by either anti-Müllerian hormone (AMH) >4 ng/mL or an abnormal ovarian size.

As part of the work up, these horses were examined for the presence of orthopedic, alimentary, vaginal or uterine pathology, and nonsteroidal anti-inflammatory drug (NSAID) (Flunixin-meglumine 1.65 mg/kg PO SID, 4 h before riding three consecutive days) treatment was used in an attempt to rule out undetected musculoskeletal related pain. If no extra-ovarian pathology or pain was found/identified, hormone modulating therapy (0.044 mg Altrenogest/kg SID PO, minimum 10 consecutive days) was used to suppress oestrus behaviour. If there was also little or no effect of the hormone modulating therapy, the horse would be a candidate for OVE. Horses also considered for OVE were mares diagnosed with an ovarian tumour based on elevated AMH or clinical signs.

Standing OVE was conducted by bilateral incision in the paralumbar fossa with ovarian pedicle ligature before excision. The bilateral procedure as standard is not considered to be lege artis when treating mares with neoplasia, hence they normally are unilateral. Due to numerous cases being without a definitive diagnosis, the bilateral removal of ovaries was conducted to eliminate any ovarian hormonal influence post-OVE. The surgical procedure was conducted by the same surgeon for the entire case series. The ovaries were submitted for histopathological examination conducted by Finn Pathologists (https://www.finnpathologists.co.uk).

A paired T-test and descriptive statistics were conducted for the entire case series regarding rideability and behaviour before and after surgery. The cases series were grouped in two categories based on the histopathologic examination/findings. One group with neoplastic findings and a group of normal findings/no macro- or microscopic indications of neoplasia. Descriptive statistics and a nested T-test was performed.

## 3. Results

The semi-quantitative result for rideability and behaviour before and after OVE is depicted in Figure 1A,B, respectively. The mean value for rideability pre- OVE was 2.3 (SD ± 2.0) compared to a mean of 6.8 (SD ± 3.4) post- OVE (*p* ≤ 0.0001). The mean value for behaviour pre- OVE was 5.3 (SD ± 3.5) and post- OVE was 8.4 (SD ± 2.7) (*p* = 0.0002).

Figure 2A,B illustrate the results following a histopathologic subgrouping. Fourteen horses out of 28 (50%) were diagnosed with ovarian neoplasia on histology. Of these, 79% percent (11/14) were categorized as granulosa-theca cell tumours, two as bilateral ovarian leiomyomas and one was a luteoma. Ten (10/28; 35.7%) mares were categorized as having normal ovaries. No histologic examination was conducted on the ovaries of the remaining four mares, hence they were excluded.

The mean value for rideability of the mares with histopathologically normal ovaries pre-OVE was 1.7 (SD ± 0.8) and post-OVE was 7.4 (SD ± 3.0). In comparison, the group with neoplastic changes had a mean rideability score pre-OVE of 2.9 (SD ± 2.6) and post-OVE of 7.0 (SD ± 3.3). In both histopathologic subdivisions, a significant difference between the mean pre- and post-OVE score was observed (*p* ≤ 0.0001) (Figure 2A). No significant difference in rideability was identified before and after OVE between the group diagnosed with or without neoplastic ovaries (*p* = 0.92, nested T-test).

Figure 2B depicts the results for behaviour pre- and post-OVE. The mean for the mares with normal ovaries pre-OVE was 5.6 (SD ± 4.1) and post-OVE was 8.5 (SD ± 2.8). In the group of mares with neoplasia, the behavioural mean pre-OVE was 4.7 (SD ± 3.4) and post-OVE was 8.6 (SD ± 2.2). Owner ratings of mare behaviour were significantly higher pre- and post-OVE in both sub-groups (*p* = 0.006). No difference in change in behaviour score was identified between the group with normal ovaries and neoplastic ovaries (*p* = 0.89, nested T-test).

Table 1, Table 2 and Table 3 depict the descriptive data of the case series. Table 1 shows the histopathologic findings. Granulosa theca cell tumours were most common, but a low number of less frequent tumours were also recorded, e.g., leiomyoma and luteoma. Table 2 shows the mean age, duration of symptoms, rideability and behaviour scores in mares with and without neoplasia. The table shows homogeneity between the two groups when comparing the listed parameters. Table 3 and Table 4 illustrate the change in rideability score after bilateral ovariectomy in mares with and without ovarian neoplasia. Following ovariectomy, rideability improved with a score ≥5 in 80% (8/10) of mares with normal ovaries and in 57% (8/14) of mares with ovarian neoplasm. A behavioural improvement of ≥5 was observed in 40% (4/10) of mares with normal ovaries, and in 43% (6/14) of mares with ovarian neoplasm.

## 4. Discussion

Bilateral ovariectomy of mares with unwanted behaviour caused a significant improvement in rideability and behaviour as perceived by the owner. Previous reports on the effect of OVE described an improvement in unwanted behaviour in relation to oestrus [1,2,3]. In the present study, the unwanted behaviour was not obviously related to the oestrus cycle. Kamm and Hendrickson (2007) ovariectomized mares with only behavioural problems, apparently not related to the oestrus cycle, and found an improvement in behaviour in 83% (19/23) of mares. In the present study, 80% (8/10) of mares with normal ovaries and rideability problems, and 40% (4/10) of mares with behavioural problems on the ground, improved following OVE. There is no obvious explanation as to why the mares with normal ovaries benefited from an OVE. Whether there was an undetected pathological problem present in the ovaries cannot be ruled out; however, histopathology is often used as golden standard to identify pathology. Another hypothesis is that some mares may benefit from an OVE in the same manner as stallions to reduce hormone-driven unfavourable behaviour and rideability following orchidectomy.

To our surprise, no observable statistical difference in rideability or behaviour following OVE could be identified between mares with or without ovarian neoplasia. It is generally accepted that mares with neoplasia, e.g., granulosa-theca cell tumours and bilateral leiomyomas, will benefit from OVE [5], but in the present study, the mares with normal ovaries equally improved in behaviour. Sixty-four percent (9/14) of the mares had granulosa theca cell tumours, which is consistent with granulosa theca cell tumours being the most frequent type of ovarian tumour in the mare [6]. The histopathologic diagnosis of the ovarian tumours presented some rare cases. Bilateral granulosa theca cell tumours have previously been described [7,8]. In a report by Sherlock et al. (2016), 25% of 52 horses had histopathologically confirmed bilateral granulosa theca cell tumours [9]. Bilateral ovarian leiomyomas have previously been described [5]. The diagnostic laboratory diagnosed a luteoma in one mare. It was not possible to find previous descriptions of luteomas in mares. Luteomas have been described in humans during pregnancy and are considered not to be neoplastic [10].

The bilateral OVE procedure is not considered to be lege artis when treating mares with ovarian neoplasia because they most frequently are unilateral. Due to numerous cases being without a definitive diagnosis before the surgical procedure the bilateral removal of ovaries was conducted in the present case series to eliminate any ovarian hormonal influence post-OVE.

The semi-quantitative scoring system based on owner observation was used by the same owner/rider before and after OVE for all twenty-eight cases. The scoring system is not objective and comparable from one horse to another, but consistent regarding the evaluating party, and accordingly the results before and after OVE for each case are comparable. Furthermore, the scores are based on multiple observations from daily handling and riding, which reduces the risk of falsely improved evaluation. A more systematic and comparable scoring system could be implemented if a professional rider was used before and after OVE. The use of one professional rider/handler would enable the scores to be compared across the case series and at the same time diminish the risk of no improvement due to unprofessional handling/riding. A behavioural assessment by a certified veterinarian specialized in behaviour before and after OVE would also be optimal, but often not realistic in a practical setting. The cut-off value for success was set as a difference of five increments/points based on a clinical demand for a substantial improvement after the surgical procedure. It could be argued that this rather conservative judgement underestimates our success rate.

The study was performed in a private practice setting, enrolling six different clinicians for the diagnostic work up. Differences in diagnostic approaches among clinicians caused missing values in the dataset, as not all diagnostic tests were conducted on all mares. The variable duration post-OVE for follow up could introduce error due to memory recall. The owner’s perception of a change in the mare’s behaviour and rideability could also be a placebo effect due to the assumption or hope that the operation will work. Two mares underwent OVE despite an inapt clinical presentation. The OVEs were conducted as a last resort before euthanasia, but neither cases benefited from the OVE. One of the mares was euthanized and a brain tumour was diagnosed at the post mortem examination In order to obtain a positive response to OVE among the mares with normal ovaries and behavioural problems, it is important to do a thorough diagnostic workup to rule out extra-ovarian pathology.

Future studies should be prospective with a diagnostic protocol suitable for a practice setting should be strictly followed by the group of clinicians. Owner assessment before and after surgery should be included, but contributions from professional riders/behaviour specialists should be considered.

## 5. Conclusions

In conclusion, a significant improvement was observed in rideability and behaviour post- ovariectomy, but no statistical difference in improvement after ovariectomy between mares with ovarian neoplasia and mares with histopathologic normal ovaries was observed. The results suggest that mares with and without neoplasia can equally benefit from ovariectomy to improve behaviour and rideability.

Despite the significant improvement observed in the present study, further research is necessary to confirm whether mares with unwanted behaviour not obviously related to the oestrus cycle and to painful conditions may benefit from ovariectomy to alter their behaviour and rideability.

## Figures and Tables

**Figure 1 animals-10-01210-f001:**
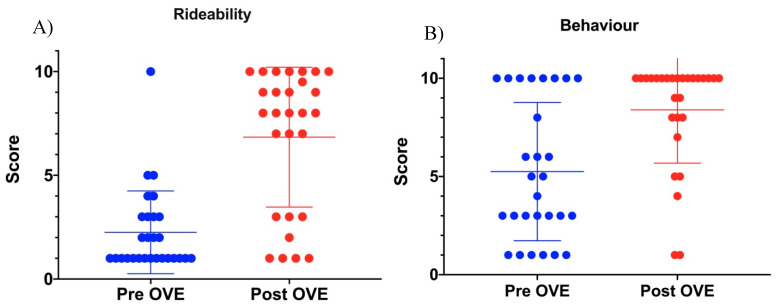
Comparison of (**A**) rideability and (**B**) behaviour pre- and post- ovariectomy (OVE). The score on the *y*-axis indicates the owners’ observation (1–10, with 10 being optimal rideability/behaviour). Each dot represents a case. Horizontal markers indicate the mean value and standard deviation.

**Figure 2 animals-10-01210-f002:**
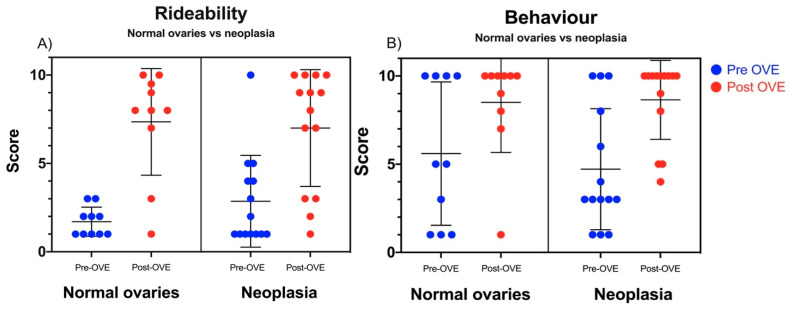
Comparison of (**A**) rideability and (**B**) behaviour pre- and post–ovariectomy (OVE) in mares with histologically normal ovaries and neoplasia. The scores on the *y*-axis indicate the owners’ observations. Horizontal markers indicate the mean value and standard deviation. (1–10, with 10 being optimal rideability/behaviour). Both rideability and behaviour were significantly improved following OVE, however no difference was noted when comparing horses with or without ovarian neoplasia.

**Table 1 animals-10-01210-t001:** Distribution of histopathologic findings.

Histological Diagnosis	Number of Mares
Normal ovaries	10
No histopathologic examination	4
Unilateral Granulosa theca cell tumor	8
Bilateral Granulosa theca cell tumor	1
Bilateral leiomyoma	3
Luteoma	1
Unspecified tumor	1
Total	28

**Table 2 animals-10-01210-t002:** Age, duration of symptoms, rideability and behaviour scores in mares with and without neoplasia. Four cases were excluded due to no histopathologic examination.

Distribution of Case Series	Normal Ovaries	Neoplasia
n	10	14
Mean age (years)	8.1 (±3.0)	9.9 (±2.9)
Mean duration of symptoms (years)	2.3 (±1.9)	1.5 (1.0)
Mean rideability score (pre-/post OVE)	1.7 (±0.8)/7.4 (±3.0)	2.9 (±2.6)/7.0 (±3.3)
Mean behaviour score (pre-/post OVE)	5.6 (±4.1)/8.5 (2.8)	4.7 (±3.4)/8.6 (±2.2)

**Table 3 animals-10-01210-t003:** Change in rideability score after bilateral ovariectomy in mares with and without ovarian neoplasia. A change in score of ≥ 5 was considered a marked positive change in behaviour following ovariectomy. Four horses were excluded due to no histologic examination.

Change in Rideability Score	Mares with Normal Ovaries (n)	Mares with Ovarian Neoplasia (n)
≥5	8	8
<5	2	6
Rate of mares with a change of ≥ 5 in rideability score	8/10 = 80%	8/14 ≅ 57%

**Table 4 animals-10-01210-t004:** Change in behaviour score after bilateral ovariectomy in mares with and without ovarian neoplasia. A change in score of ≥ 5 was considered a marked positive change in behaviour following ovariectomy. Four horses were excluded due to no histologic examination.

Change in Behaviour Score	Mares with Normal Ovaries (n)	Mares with Ovarian Neoplasia (n)
≥5	4	6
<5	6	8
Rates	4/10 = 40%	6/14 ≅ 43%

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
