# Peer review of "Moody Mares—Is Ovariectomy a Solution?"

_animals, 2020, doi:10.3390/ani10071210_

Round 1
Reviewer 1 Report
REVIEW ANIMALS: Moody mares – Is ovariectomy a solution?
This is an interesting study and it is pleasing to see case study series coming forward for publication and integrating statistical analyses within the evaluation of the cases included. Behavioural issues in mares are a common problem and this paper will be of interest to many readers, and contributes to this field.
Line 3 – assume this is the running title?
Affiliations: please provide these in full as per journal guidance
Simple summary:
Sets scene and provides information about methods, but should summarise the entirety of your study therefore results linked to aim of evaluating behaviour post ovarioectomy should be included.
Line 10: Please add simple to the title here
Line 13: as the summary is aimed at the public would be beneficial to add (cancers) after neoplasms
Line 17 and 18: suggest removing n=
Line 18-19: advise amending to ‘the ovaries of the remaining 4 mares were not examined to identify if cancer was present’. for increased clarity for this audience
Abstract:
Generally a good summary of the study provided, inferences made require some amendment as outlined below.
Lines 30-31: where you have provided (4/10) I would advise tweaking presentation as follows: 40% (4/10)
Introduction:
Good background linking oestrus cycle and behavioural issues that can present in the mare and routine treatment approaches.
Line 64-65: this sentence (The present study…) feels out of place here, why is this approach taken or should it be that you are describing a novel approach or similar – requires editing to make purpose clearer
Line 72-74: aim presented here does not match that within simple summary and abstract of comparing post OVE behaviour across the two groups of mares assessed, please amend to ensure all match. Personally I would aim to compare the behaviour between groups
Line 75-76: suggest amending to ‘neoplasia would have a more disagreeable demeanour pre-OVE than mares without neoplasia, and therefore the former group would have a higher…’
Materials and methods:
Generally clear but how rideability was defined should be included. It would also be beneficial to provide an overview of the signalment of the mares within the two groups e.g. age, breed, discipline, experience level. The variable durations post-OVE for follow up could introduce error due to memory recall or potentially due to a placebo effect, would be good to consider this either here or within discussion as a potential limitation. Also suggest including details that median and IQR ranges were calculated for each group. Please also include a statement re ethical approval for the study.
Line 93: remove furthermore don’t think required here
Results:
Scope to increase impact by adapting structure and flow to optimise presentation to reader.
Line 114-117: I am not a big fan of presenting what tables include and then listing them as I feel this leaves the reader to make their own interpretation and represents a missed opportunity for the author/s to highlight the key findings from their perspective that they want the reader to take notice of- I would suggest amending to take this approach here and to highlight changes pre and post OVE in behaviour and rideability, you do this in line 129-131, I would start with this and then link to tables and line 128-129 / figures after this to improve synthesis and flow
Table 2 – rideability and behaviour should be presented as median±IQR as these measures are categorical due to rating applied – could give both mean and median here
Figures 1 and 2: include full explanation of OVE in legend
Line 150: suggest amending to Owner ratings of mare behaviour was significantly higher pre- and post-OVE in both sub groups (p = 0.006). Then I would report the median values afterwards.
Discussion:
Good consideration of the results obtained from a practical and clinical perspective, with reference to relevant literature.
Please include a limitations paragraph / sub section to consider some of the potential areas which could influence your results such as the duration in follow up time as identified above.
Conclusions:
Line 216-217: rather than stating there was a difference, reword to state what the difference was i.e. increased / decreased as more informative
Reviewer 2 Report
The paper aims to investigate whether ovariectomy is successful in reducing unwanted behavior and rideability in problem mares.
The subject is of interest and the paper is well written, however some limitation exists about the system for evaluating the behavior and rideability of the mares. Owner's opinion about its mare behavior and rideability could be affected by its expectations following ovariectomy. A more objective method would have provided a higher scientific value. Anyway, the authors report the limits of the study in the discussion section.
In the reviewer's opinion the questionnaire form provided to the owners should be included in the paper.
Minor comments:
line 31-33: the sentence is not clear, please rephrase it.
line 66: please change "the mares participating was" with "the mares participating were"
line 218-220:I suggest yo modify the sentence as follows: "Despite the significant improvement observed in the present study, further research is necessary to confirm whether mares with unwanted behavior not obviously related to the oestrus cycle and to painful conditions may benefit from ovariectomy to alter their behavior and rideability.
